# Investigating the Effect of Ni Loading on the Performance of Yttria-Stabilised Zirconia Supported Ni Catalyst during CO₂ Methanation

Osaze Omoregbe [1,*], Artur J. Majewski [1], Robert Steinberger-Wilckens [1] and Ahmad El-kharouf [2]

1 Centre for Fuel Cell and Hydrogen Research, School of Chemical Engineering, University of Birmingham, Edgbaston, Birmingham B15 2TT, UK
2 Ove Arup & Partners Ltd., 3 St Paul's Place, Norfolk Street, Sheffield S1 2JE, UK
* Correspondence: oxo850@bham.ac.uk

**Abstract:** CO₂ methanation was studied on Ni-based yttria-stabilised zirconia (Ni/YSZ) catalysts. The catalysts were prepared by the wet impregnation method, where the amount of Ni content was varied from 5% to 75%. Thereafter, the prepared catalysts were analysed by BET, XRD, SEM and H₂-TPR. BET results showed an initial increase in the surface area with an increase in Ni loading, then a decrease after 30% Ni loading. The XRD results revealed that the Ni crystallite size increased as the Ni loading increased, while the H₂-TPR showed a shift in reduction peak temperature to a higher temperature, indicating that the reducibility of the catalysts decreased as the Ni loading increased. The activity of the synthesised catalysts for CO₂ methanation was studied by passing a mixture of H₂, CO₂ and N₂ with a total flow of 135 mL min⁻¹ and GHSV of 40,500 mL h⁻¹ g⁻¹ through a continuous flow quartz tube fixed-bed reactor (I.D. = 5.5 mm, wall thickness = 2 mm) containing 200 mg of the catalyst at a temperature range of 473 to 703 K under atmospheric pressure and a H₂:CO₂ ratio of 4. The tested Ni/YSZ catalysts showed an improvement in activity as the reaction temperature increased from 473 K to around 613 to 653 K, depending on the Ni loading. Beyond the optimum temperature, the catalyst's activity started to decline, irrespective of the Ni loading. In particular, the 40% Ni/YSZ catalyst displayed the best performance, followed by the 30% Ni/YSZ catalyst. The improved activity at high Ni loading (40% Ni) was attributed to the increase in hydrogen coverage and improved site for both H₂ and CO₂ adsorption and activation.

**Keywords:** power-to-gas; CO₂ methanation; YSZ; Ni loading; reducibility





## 1. Introduction

The global energy demand continues to increase as a response to the growing world population and advancement in technology. Currently, fossil fuels account for the largest percentage of total energy generation. In 2020, fossil fuel sources provided over 80% of the global energy demand [1]. However, there are growing concerns due to the associated greenhouse emissions, which are largely responsible for the climate change crisis. The worldwide average amount of CO₂ in the atmosphere reportedly hit a new record high of 413.2 ppm in 2020 [2]. This new level of CO₂ has further placed pressure on nations to act fast in reducing greenhouse gas emissions. This was also the main topic of discussion in the COP26 and COP27, the United Nations Climate Change Summits held between 31 October to 12 November 2021 and 6 to 18 November 2022, respectively. Several solutions are being proposed, including the planting of trees and the replacement of fossil fuels with renewable and clean sources. However, the total energy produced from renewable sources is not yet able to meet the current world energy demand. Other problems associated with most renewable sources include energy storage challenges and volatility (weather and season dependency). Although great efforts have been channelled toward scaling up renewable energy production and energy storage capacity, the use of fossil fuels cannot be phased out

in the short term. The international energy outlook data from the U.S. Energy Information Administration [1] showed that fossil fuels and their infrastructure will still be relevant and highly consumed in the next few decades.

Therefore, it is important to find ways of minimising $CO_2$ emissions during the use of fossil fuels whilst the transition to renewable sources continues. One way of achieving this is to capture the $CO_2$ emitted during the combustion process and convert it to useful products. This is a feasible process because $CO_2$ is an essential feedstock for the production of C1-based fuels and their derivatives via various processes [3]. In recent times, the hydrogenation of $CO_2$ into synthetic natural gas (SNG) has become a very attractive means of storing the excess energy generated from renewable sources [4,5]. Generally, energy from a renewable source is used to power electrolysers for hydrogen production, which is then utilised in $CO_2$ hydrogenation through the Sabatier reaction to form synthetic methane (Equation (1)). The advantage of SNG over hydrogen is that it is fully compatible with the existing natural gas infrastructure. This power-to-gas strategy helps to minimise $CO_2$ emissions while solving the storage challenge of renewable energy. $CO_2$ captured from fossil fuel processes will in this way be re-utilised, thus reducing fossil $CO_2$ emissions, but not preventing them. If the $CO_2$ is obtained from biomass sources, no fossil $CO_2$ will be released at all. Besides the climate mitigation point of view, it is also believed that $CO_2$ methanation will be economically competitive if carbon capture and the electrolytic process for the required $H_2$ are further improved and reduced in cost. As electricity from renewable energy sources has become the cheapest type of electricity in many parts of the world, and with the current developments in natural gas market, favourable economic conditions are clearly visible [6].

$$CO_2 + 4H_2 \rightleftharpoons CH_4 + 2H_2O \quad \Delta H_{298K} = -164 \text{ kJ mol}^{-1} \tag{1}$$

Despite the promising potential of $CO_2$ hydrogenation for renewable energy storage and $CO_2$ utilisation, technological challenges remain, including heat management and catalyst deactivation. $CO_2$ hydrogenation is thermodynamically favoured at low temperatures [7,8] but the temperature in the reactor is likely to increase above the starting temperature due to the strong exothermal character of the reaction. Consequently, reactor overheating can lead to hot spot formation which again causes catalyst deactivation through agglomeration and carbon deposition [9,10]. Interestingly, the heat released in the exothermic $CO_2$ hydrogenation reaction can provide the energy required for the adsorption of $CO_2$ on a dispersed adsorbent to spill over to the catalyst active sites where it is further converted [11]. Therefore, heat management is essential in the $CO_2$ hydrogenation system for process optimisation.

It is also important to develop thermally stable catalysts with high activity and exceptional resistance to carbon deposition and sintering at the $CO_2$ hydrogenation reaction conditions. In this regard, extensive research has been conducted on the application of various kinds of catalysts for the $CO_2$ hydrogenation reaction [12–14]. Group VIII metals such as Pt, Pd, Ru, Rh, and Ni are among the most widely studied metal catalysts for $CO_2$ hydrogenation owing to their high activity toward $CO_2$ conversion [15–19]. Although the noble metal catalysts (Pt, Pd, Ru, Rh) have superior catalytic activity and stability compared to Ni, their application on a large scale is not feasible due to their high cost [20]. Hence, for $CO_2$ hydrogenation, Ni-based catalysts are considered suitable alternatives to noble metal catalysts since they are considerably active, inexpensive and readily available [14]. Nevertheless, Ni-based catalysts are easily deactivated due to their poor resistance to carbon formation and to sintering during $CO_2$ hydrogenation reactions [14]. Therefore, Ni catalysts are modified by the synthesis method [21,22], by application on supports with a good surface for the dispersion of the active metal [23], or by the incorporation of a promoter into the catalyst framework to enhance the dispersion of the active metal and improve $CO_2$ adsorption, whilst minimising sintering [24].

Supports such as alumina ($Al_2O_3$), ceria ($CeO_2$), silica ($SiO_2$), titania ($TiO_2$), and zirconia ($ZrO_2$) have been extensively employed in the modification of Ni-based catalysts

for $CO_2$ hydrogenation reactions due to their ability to enhance the activity of Ni [25–27]. Specifically, $ZrO_2$ has shown to be exceptional catalyst support for $CO_2$ hydrogenation owing to its high thermal and chemical stability, high mechanical strength, and strong resistance to carbon deposition. These superior properties of $ZrO_2$ can be attributed to its physicochemical properties which include the presence of defect sites (e.g., oxygen vacancies), Lewis acid sites ($Zr^{3+}$, $Zr^{4+}$), adsorbed $O^{2-}$ and OH groups (basic sites), and $Zr^{4+}$-$O^{2-}$ acid-base pairs [28,29]. These properties also make the tuning and doping of $ZrO_2$ more feasible for various catalytic processes. Moreover, the Lewis basic oxygen vacancies on $ZrO_2$ are crucial for the adsorption and activation of $CO_2$ [30] which is one of the major steps in the mechanism of $CH_4$ formation during $CO_2$ hydrogenation.

Generally, pure zirconia exists in three different crystal configurations, at atmospheric pressure and different temperatures: monoclinic (m-$ZrO_2$), tetragonal (t-$ZrO_2$) and cubic (c-$ZrO_2$) [31–35]. Notwithstanding the great properties displayed by $ZrO_2$, its redox activity is low due to the difficulty of its surfaces in discharging lattice oxygen atoms to form vacancies [36]. This challenge is overcome by doping the $ZrO_2$ with metals with a lower valency than Zr (less than +4) such as Ni, La, Co, Ca, and Ti. The introduction of these dopants helps lower the energy required for the formation of oxygen vacancies through the substitution of some of the $Zr^{4+}$ ions and the cleavage of the local symmetry of the lattice structure [36]. The introduction of lower valence dopants results in the alteration of the electronic arrangement of $ZrO_2$, which affects the catalytic properties. It is worth noting that two unpaired electrons are left behind when oxygen vacancies are formed. These unpaired electrons can influence the catalytic cycle by enhancing the catalyst's surface activity [36].

At low temperatures, the alteration in the crystal structure of $ZrO_2$ can cause instability which limits its use in the industry [37,38]. This is why the cubic and tetragonal forms are generally stabilised at room temperature by doping with various metal oxides such as $CeO_2$, MgO, CaO and $Y_2O_3$ [38]. Consequently, $Y_2O_3$ is mostly employed in stabilising $ZrO_2$ owing to the improved stability effect it provides [32]. Another reason for the high preference for stabilising $ZrO_2$ by $Y_2O_3$ is its influence in creating an oxygen vacancy in the anionic sub-lattice when two $Zr^{4+}$ ions are replaced by two $Y^{3+}$ ions, and this assists in the migration of oxygen ions through the yttria-stabilised zirconia (YSZ) material [39,40].

The use of $ZrO_2$ as a support for $CO_2$ methanation catalysts has been reported by many researchers. Gac et al. [41] studied the effect of Ni loading on $Al_2O_3$, $ZrO_2$ and $CeO_2$ supports. The authors reported that the increase in Ni loading from 10% to 40% on the $ZrO_2$ resulted in a gradual decrease in the Ni crystallite size. The $CeO_2$-supported catalysts showed the best performance followed by the $ZrO_2$-supported catalysts in terms of $CO_2$ conversion at 553 K. An increase in Ni loading was favoured for the $CeO_2$ support, the performance of the $ZrO_2$-supported catalyst was found to decrease after 20% Ni loading. In another study by Traitangwong et al. [42], Ni loading varied from 15% to 45% over Ni-modified ceria-zirconia support (e.g., $Ni_{0.05}Ce_{0.20}Zr_{0.75}O_2$) and the catalysts were tested for $CO_2$ methanation. The catalyst with 45% Ni loading exhibited the best performance. The authors stated that increasing the Ni loading favoured the number of $H_2$ molecules that were activated, which translated into higher catalytic performance.

Kosaka et al. [43] investigated the influence of Ni content on the performance of a Ni/YSZ catalyst by varying the Ni loading from 25% to 75%. The authors found that the catalyst performance was favoured with the increase in the Ni content where the 75% Ni/YSZ catalyst displayed the best activity. Kesavan et al. [44] investigated the influence of Ni particle size on the performances of Ni-based YSZ-supported catalysts during $CO_2$ methanation. Variation in $Ni^0$ particle sizes for the Ni/YSZ catalysts was achieved by employing different preparation steps, namely wetness impregnation, electroless plating and mechanical mixing. The study revealed that the size of $Ni^0$ and its morphology influenced the Ni/YSZ catalysts, where the catalyst with a smaller $Ni^0$ particle size exhibited the best performance. It should be noted that $Ni^0$ (reduced nickel) is obtained when nickel oxide (NiO) is reduced in a hydrogen environment at the reduction temperature which depends on the catalyst structure.

Motivated by these experimental findings, the current study aims to investigate the effect of varying the Ni content on the performance of YSZ catalysts in $CO_2$ methanation by examining a wide range of Ni loading. The Ni-based YSZ catalysts were prepared by wet impregnation to obtain 5%, 10%, 20, 30%, 40, 50% and 75% Ni loading, and then tested for the $CO_2$ methanation reaction at different temperatures.

## 2. Results and Discussion

### 2.1. Catalyst Characterisation

2.1.1. Textural Properties

The $N_2$-adsorption/desorption properties of YSZ and the freshly prepared Ni/YSZ catalysts were investigated. Their adsorption curves are presented in Figure 1. As seen in the figure, the adsorption curves of all the samples developed into type-IV isotherms with an H3-type hysteresis loop based on the IUPAC classifications [45,46]. This kind of hysteresis loop is caused by capillary condensation and evaporation at high relative pressures within the range of 0.8 to 1. This $N_2$-adsorption/desorption property is typical for mesoporous materials consisting of slit-shaped pores [47]. In addition, the similarity in the $N_2$-adsorption/desorption isotherms of both the YSZ and Ni/YSZ catalysts revealed that there was no significant alteration in the YSZ support's framework after the impregnation of Ni. Notwithstanding, there were changes in the BET surface area, pore size and pore volume after the impregnation process as shown in Table 1. The BET surface area initially decreased with increasing Ni content from 5% to 10% and then increased until 30% Ni loading then declined again with high Ni loading of above 40%. The observed decrease in the surface area can be attributed to the partial blockage of the pores due to the dispersion of Ni in the support. Furthermore, the average pore size and pore volume relatively increased with higher Ni loading until 30% Ni loading and decreased above 30% Ni loading.

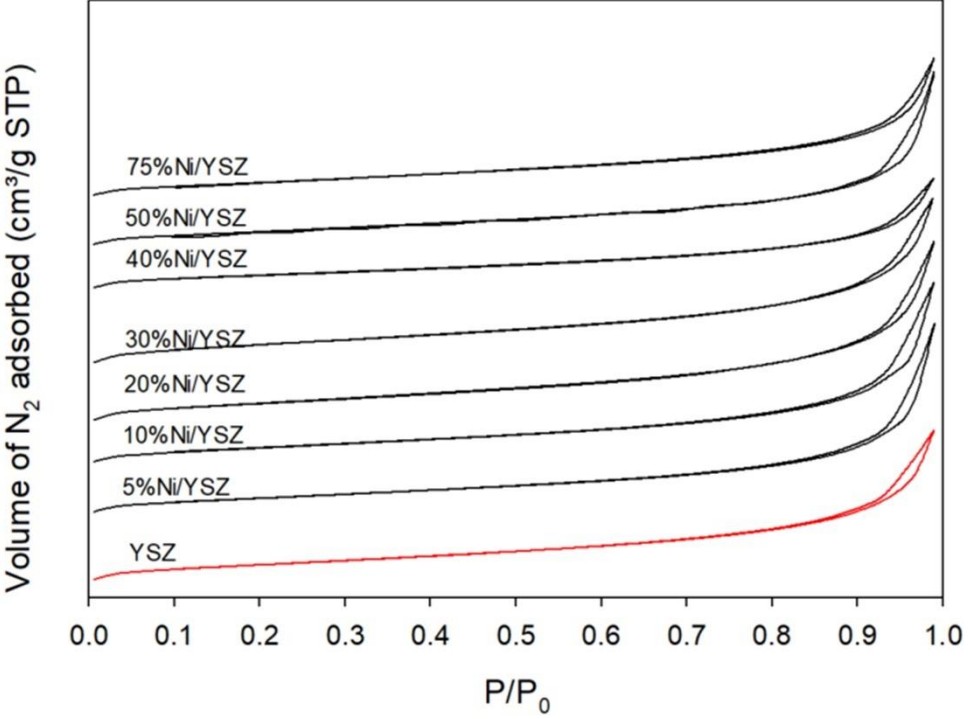

**Figure 1.** $N_2$-adsorption/desorption isotherms of the YSZ support and the calcined Ni/YSZ catalysts before the reaction.

**Table 1.** Physicochemical properties of the YSZ support and the calcined Ni/YSZ catalysts before the reaction.

| Samples | BET Surface Area (m² g⁻¹) | Pore Size (nm) [a] | Pore Volume (cm³ g⁻¹) [b] | Crystallite Size (nm) [c] | |
|---|---|---|---|---|---|
| | | | | $d_{NiO}$ | $d_{Ni}$ |
| YSZ | 5.825 | 5.639 | 0.0082 | - | - |
| 5% Ni/YSZ | 5.246 | 6.029 | 0.0079 | 21.73 | 25.74 |
| 10% Ni/YSZ | 5.335 | 6.356 | 0.0085 | 31.36 | 34.39 |
| 20% N/YSZ | 6.221 | 5.922 | 0.0092 | 31.24 | 36.52 |
| 30% N/YSZ | 6.081 | 5.846 | 0.0089 | 33.73 | 36.22 |
| 40% N/YSZ | 4.237 | 5.675 | 0.0060 | 35.75 | 39.11 |
| 50% Ni/YSZ | 4.842 | 6.064 | 0.0073 | 35.83 | 40.30 |
| 75% Ni/YSZ | 4.731 | 5.929 | 0.0070 | 35.92 | 41.08 |

[a] Data were obtained from the BJH adsorption average pore diameter (4 v/A). [b] Data were obtained from the BJH single point adsorption at P/Po = 0.95. [c] Estimated from Scherrer Equation with the (2 0 0) reflection for the NiO phase in the unreduced Ni/YSZ catalysts and (1 1 1) for the Ni⁰ phase in the reduced Ni/YSZ catalysts in the XRD data.

### 2.1.2. XRD Analysis

The crystalline phases present in all the catalysts were investigated by powder XRD. The recorded diffractograms of both the freshly prepared and reduced catalysts are shown in Figure 2a,b. The X-ray peaks were matched with the Joint Committee on Powder Diffraction Standards (JCPDS) database [48,49]. The NiO and Ni⁰ particle sizes were estimated using the Debye–Scherrer equation [50] (Equation (2)):

$$d_P(nm) = \frac{K\lambda}{\beta \cos\theta} \tag{2}$$

where $d_P$ is the particle size in nm, K is the shape factor with the value of 0.9 (assuming a spherical particle), $\lambda$ is the X-ray wavelength of the X-ray source Cu K$\alpha$ (1.5406 Å), $\beta$ is the line broadening at half the maximum intensity (FWHM) of the NiO or Ni⁰ diffraction peak and $\theta$ is the Bragg angle in radian.

An analysis of the patterns for the freshly prepared catalysts (Figure 2a) indicated the existence of YSZ peaks (JCPDS 81–1550) at $2\theta = 29.9°$ (1 1 1), 34.6° (2 0 0), 50.0° (2 2 0), 59.4° (2 2 2), 62.3° and 73.5° (4 0 0) [44,51]. NiO peaks (JCPDS 73–1519) were observed at $2\theta = 37.4°$, 43.4°, 75.6° and 79.5° for the 30% Ni, 50% and 75% Ni catalysts, while the NiO peak intensities observed at $2\theta = 37.4°$, 43.4° and 75.6° for the 5% Ni and 10% Ni catalysts was very weak. In addition, there was an overlap between the YSZ and NiO phases at $2\theta = 62.8°$. After catalyst reduction in $H_2$, the peak at $2\theta = 37.4°$ disappeared while the peak at 43.4° shifted to around 44.5°–45° indicating the presence of Ni⁰ metal (Figure 2b). A new peak appeared at about 52°, indicating the presence of Ni⁰ metal. Furthermore, the particle size calculation using the Scherrer Equation (Equation (5)) showed that there was a relative increase in Ni particle size after the reduction in $H_2$.

This increase in Ni particle size after the reduction of NiO was unexpected. However, the reduction of catalysts at elevated temperatures has been reported to encourage crystal growth through migration and coalescence, or by atom diffusion [52–55]. Yi et al. [56] employed the atom diffusion mechanism to conduct a sintering kinetic study for the growth of Ni during the reduction of a series of Ni-based catalysts. The authors stated from their findings that the distance between the diffused Ni species and metal-support interactions (MSI) or variation in pore size and shape influenced the order of sintering kinetics of Ni.

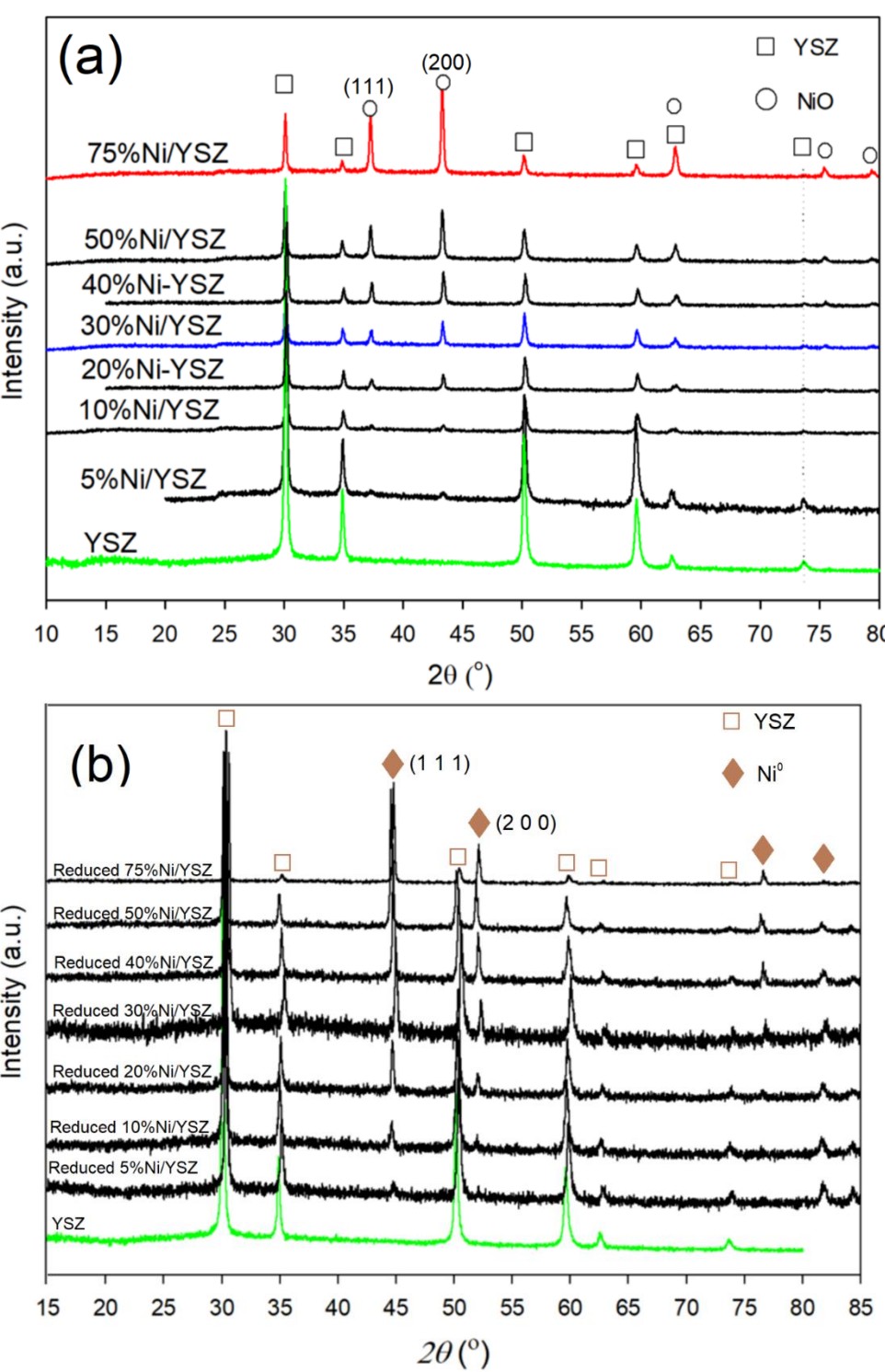

**Figure 2.** XRD patterns of (**a**) calcined and (**b**) reduced Ni/YSZ catalysts before the reaction.

Notwithstanding the occurrence of Ni crystal growth, all the reduced catalysts exhibited similar XRD patterns, which indicated that the Ni/YSZ structure was not significantly altered after the reduction process.

### 2.1.3. SEM Measurement

The SEM images of the calcined Ni/YSZ catalysts and SEM-EDS elemental mapping of the reduced catalysts are represented in Figures 3 and 4, respectively. The YSZ support (Figure 3a) showed a rough surface with uniform morphology. The SEM image of the

5% Ni/YSZ catalyst Figure 3b showed a uniform surface, which indicates an even distribution of the Ni particles. This even distribution and absence of sintering of the NiO particles were likely due to the low content of NiO particles in relation to the YSZ support's high surface availability. With increasing Ni content (Figure 3c–f), the clustering of NiO particles became evident, which was in line with the BET analyses.

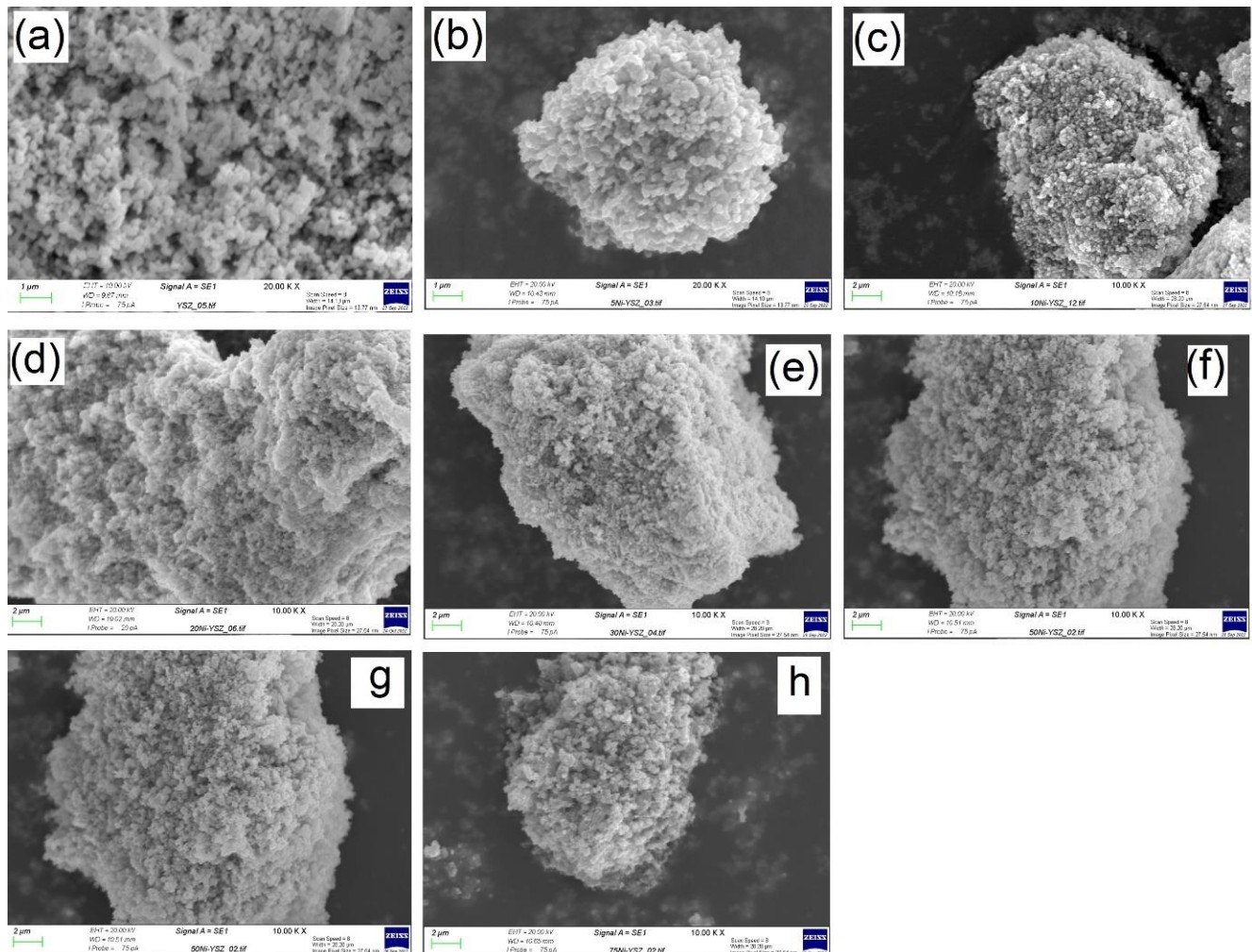

**Figure 3.** SEM image of (**a**) YSZ, (**b**) 5% Ni/YSZ, (**c**) 10% Ni/YSZ, (**d**) 20% Ni/YSZ, (**e**) 30% Ni/YSZ, (**f**) 40% Ni/YSZ, (**g**) 50% Ni/YSZ and (**h**) 75% Ni/YSZ calcined catalysts before the reaction.

The EDX mapping of the reduced catalysts shown in Figure 4 gives an insight into how Ni particles are distributed on the catalyst surfaces. The circled regions have a high concentration of Ni (clustering), as indicated by the dense green colour, while the dark spots seen in most of the EDX mapping of the catalysts are due to the cavities resulting from their rough surfaces. At 5% Ni, the Ni particles were well dispersed on the surface. The presence of Ni particles started becoming more visible as the loading increased from 10% to 75%. However, the mappings of the catalysts with higher Ni loadings from 30% to 75% showed an uneven distribution of Ni particles. In addition, the regions showing the presence of Ni clusters (dense green colour) also increased with the Ni loading. These results further support other analyses indicating the occurrence of significant sintering at higher Ni-loading.

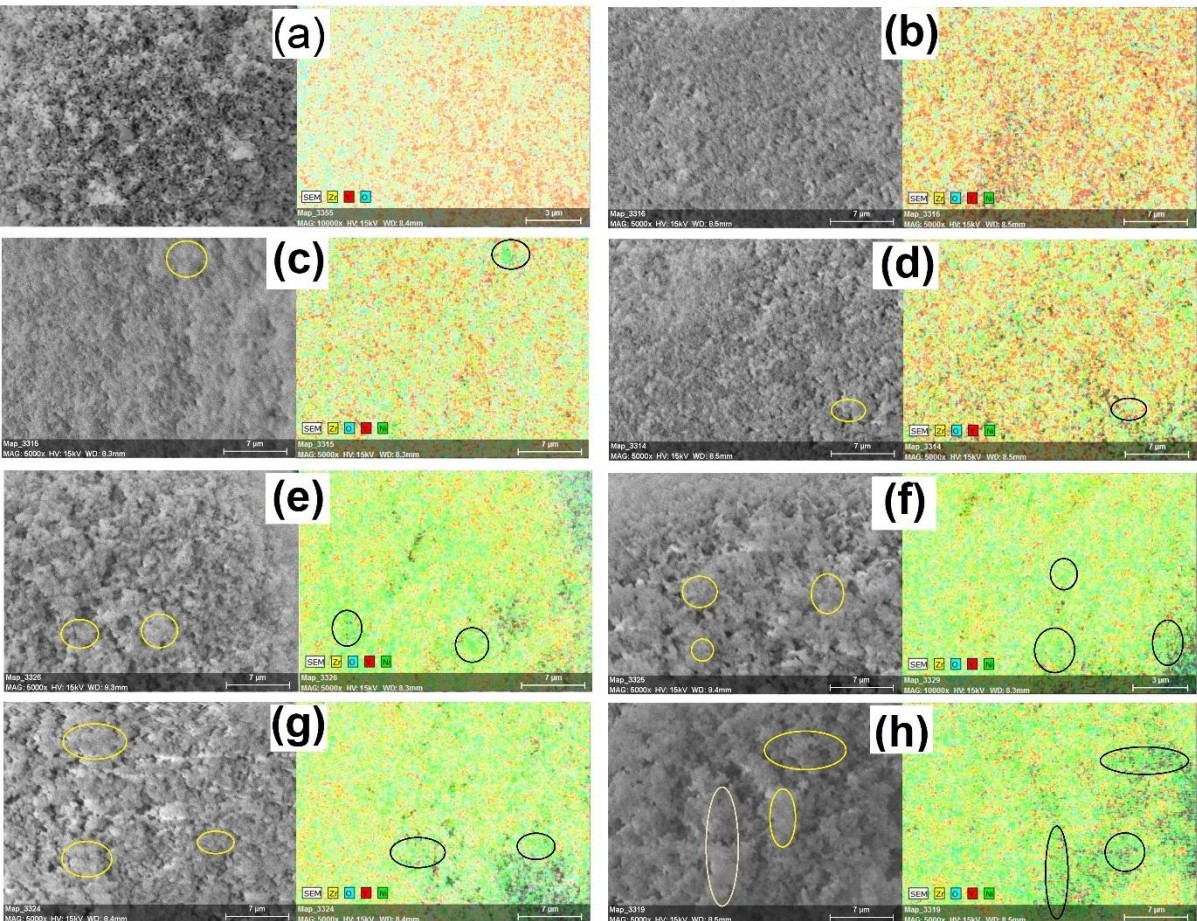

**Figure 4.** SEM-EDS image of (**a**) YSZ, (**b**) 5% Ni/YSZ, (**c**) 10% Ni/YSZ, (**d**) 20% Ni/YSZ, (**e**) 30% Ni/YSZ, (**f**) 40% Ni/YSZ, (**g**) 50% Ni/YSZ and (**h**) 75% Ni/YSZ calcined catalysts after reduction.

### 2.1.4. Catalyst Reducibility

The reduction behaviour of all the catalysts was investigated using the TPR-MS technique in the $H_2(20\%)/N_2$ environment. The TPR profiles of the catalysts are shown in Figure 5. Generally, when the reduction of a catalyst in the $H_2$ environment begins, the amount of hydrogen in the system will start dropping (increase in $H_2$ consumption). The increase in $H_2$ consumption will continue until a maximum point is reached, and then the signal will start decreasing. The temperature at which the maximum point is reached normally is used to identify the reduction temperature of a catalyst. Therefore, the reduction temperatures of the catalysts in this study were determined by the readings at the maximum peaks from the $H_2$-TPR profile. As seen in the figure, both the 5% Ni and 10% Ni loading exhibited similar behaviour at temperatures ranging from 567 to 810 K. Meanwhile the reduction of 30% Ni, 50% and 75% Ni catalysts was in the range of 520 to 860 K. The reduction peak temperatures for all the catalysts are given in Table 2. The first peak observed at lower temperatures for all the catalysts can be attributed to the reduction of bulk NiO with weak support interaction to $Ni^0$, while the second peak at higher temperatures can be ascribed to the reduction of the highly dispersed NiO particles with stronger MSI [57,58]. These results revealed that higher Ni loading of around 30–40% decreased the reducibility of the catalysts as seen in the shift in the reduction peaks to the right and enlarged peak 2, which means that a higher temperature is required for the complete reduction of the NiO particles. This indicated higher interaction with the support and was in line with the BET results. A further increase in Ni loading to 50 to 75% resulted in an increase in Ni with low MSI and a reduction of Ni at a lower temperature, as indicated by enlarged peak 1

(Figure 4). This confirmed the BET and XRD results and the formation of large Ni particles with low MSI.

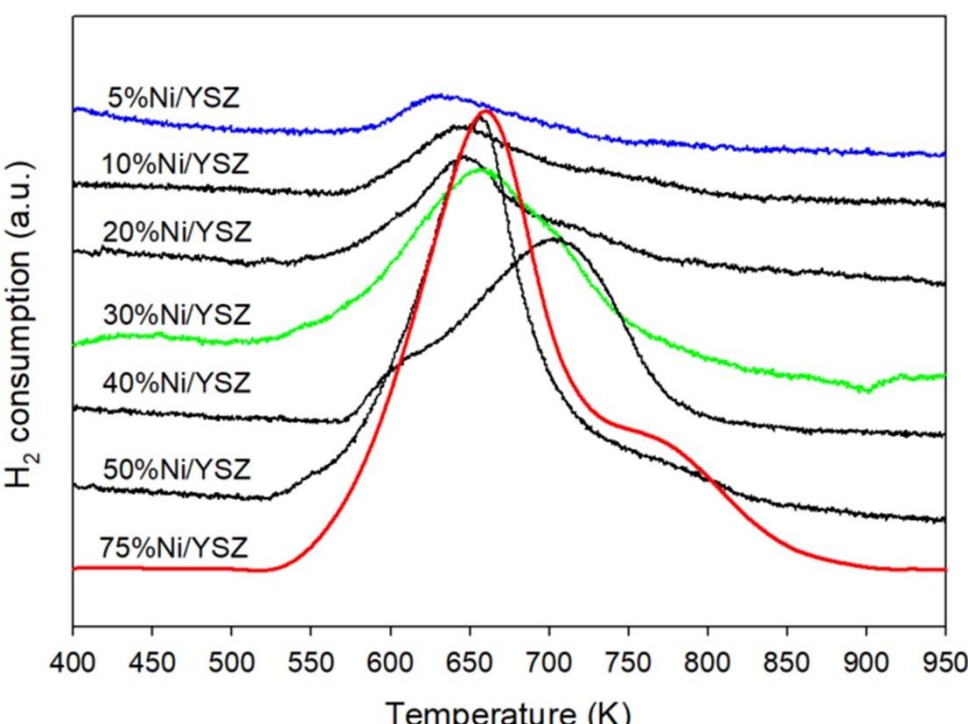

**Figure 5.** H$_2$-TPR profiles of 5% Ni/YSZ, 10% Ni/YSZ, 20% Ni/YSZ, 30% Ni/YSZ, 40% Ni/YSZ, 50% Ni/YSZ and 75% Ni/YSZ calcined catalysts before the reaction.

**Table 2.** Reduction peak temperatures during the H2-TPR of calcined Ni/YSZ catalysts.

| Samples | Reduction Peak Temperatures (K) * | |
|---|---|---|
| | **Peak 1** | **Peak 2** |
| 5% Ni/YSZ | 629 | - |
| 10% Ni/YSZ | 641 | 734 |
| 20% N/YSZ | 645 | 724 |
| 30% N/YSZ | 654 | 748 |
| 40% N/YSZ | 601 | 706 |
| 50% Ni/YSZ | 654 | 778 |
| 75% Ni/YSZ | 659 | 764 |

* Reduction peak temperatures were obtained from H$_2$-TPR profiles depicted in Figure 5.

The reduction of all the catalysts was completed before reaching 973 K. Therefore, 973 K was selected as the catalyst reduction temperature to ensure that no NiO was taking part in the reaction during the activity testing.

## 2.2. Catalyst Activity Test

The results of the activity test over the Ni/YSZ catalysts during CO$_2$ methanation are shown in Figure 6a,b. As seen in Figure 6a, there was a steady but low X$_{CO_2}$ at temperatures between 473 to 553 K. However, beyond 553 K a sharp increase in the CH$_4$ formation was observed. The increase in X$_{CO_2}$ continued with increasing temperature until around 633 K for all tested catalysts. Thereafter, the X$_{CO_2}$ declined slowly at higher temperatures. Therefore, the optimal temperature for X$_{CO_2}$ was from 613 to 653 K. All the catalysts exhibited relatively high activity at the optimal temperature. These superior properties of the catalysts are likely related to the unique physicochemical properties of the YSZ support which include the presence of defect sites such as the oxygen vacancies [28,29].

These properties aid the adsorption and activation of $CO_2$ [30] which is a crucial step in the mechanism of $CH_4$ formation during the $CO_2$ methanation reaction.

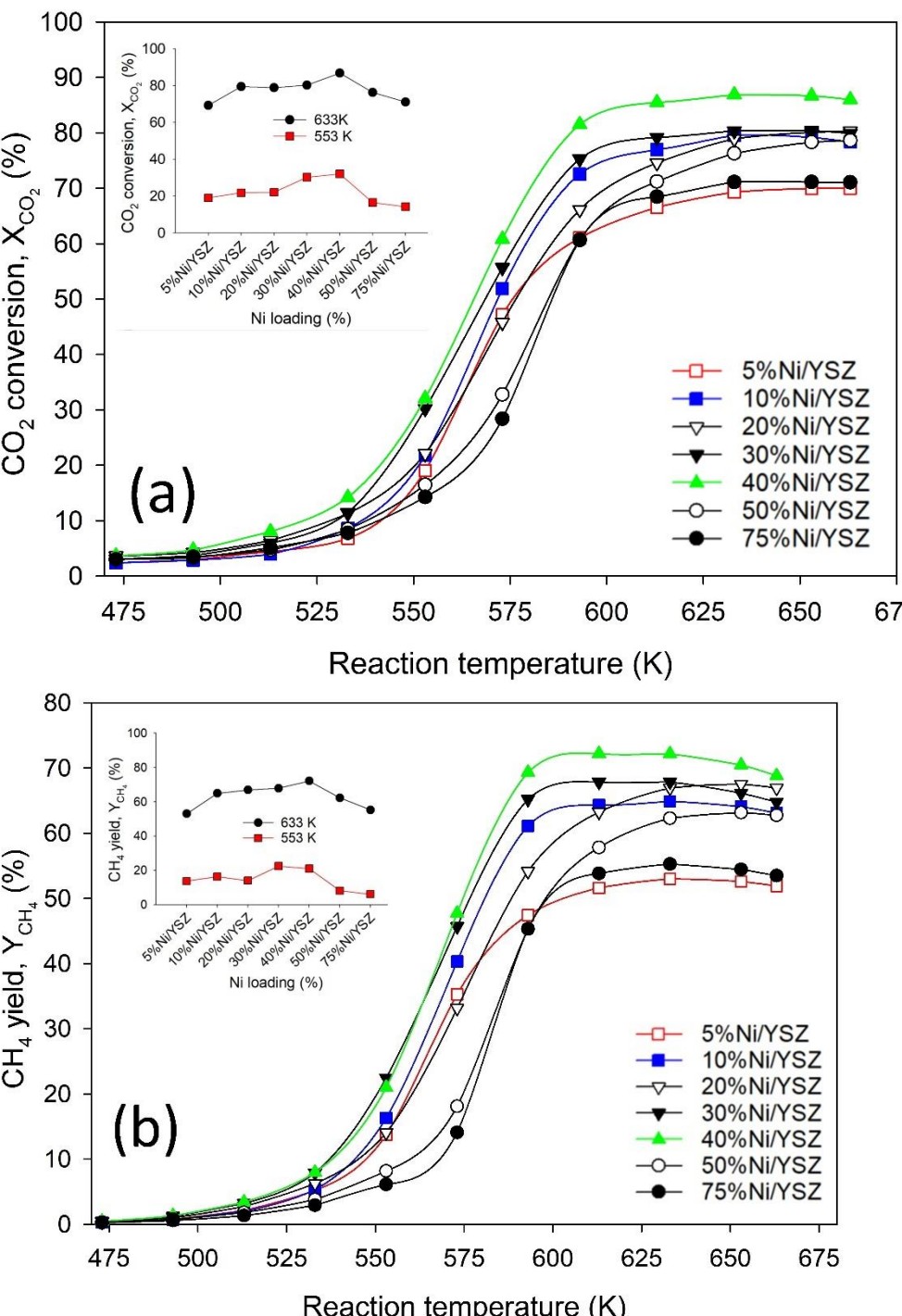

**Figure 6.** Activity test profile showing (**a**) $CO_2$ conversion and (**b**) $CH_4$ yield during $CO_2$ methanation over Ni/YSZ catalysts. Reaction conditions: Total flow = 135 mL min$^{-1}$, $H_2$:$CO_2$ ratio = 4, pressure = 1 bar, temperature = 473–703 K, GHSV 40,500 mL h$^{-1}$ g$^{-1}_{cat}$.

In Figure 6b, a similar trend was observed for $Y_{CH_4}$, where the amount of $CH_4$ produced was initially favoured as the temperature increased, and then decreased when the temperature was raised above the optimum value. It is worth noting that the $CO_2$ methanation reaction is thermodynamically feasible at low temperatures due to its exothermic nature (see Figure 7), but the results from this study revealed that high $Y_{CH_4}$ cannot be

achieved at low temperatures. This is due to the substantial kinetic limitations associated with the cleavage of $CO_2$ bonds during its reduction processes to produce other carbon species, which leads to the formation of $CH_4$ [59,60]. Mebrahtu et al. [61] acknowledged that the $CO_2$ methanation reaction is thermodynamically feasible at low temperatures, but the authors stated that high operating pressures will be required to conduct $CO_2$ methanation at low temperatures for a favourable equilibrium composition.

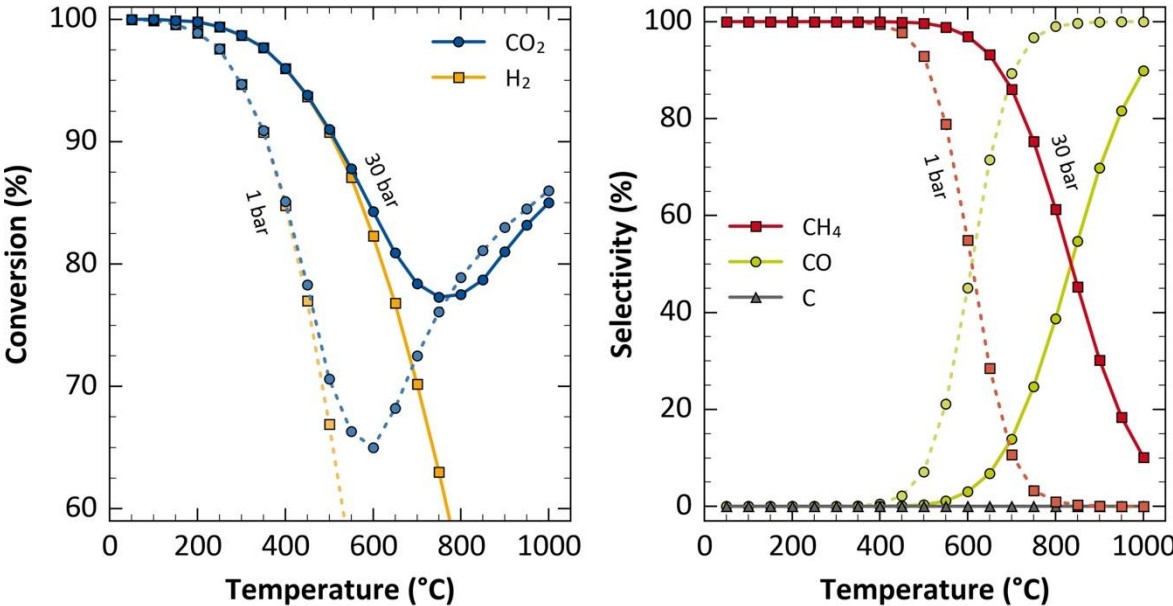

**Figure 7.** Activity profile showing the influence of temperature and pressure on $CO_2$ equilibrium conversion and selectivity for $CO_2$ methanation conducted at a $H_2$:$CO_2$ ratio of 4. Adapted with permission from ref [61]. Copyright 2019 Elsevier B.V.

On the other hand, conducting $CO_2$ methanation at elevated reaction temperatures favours side reactions such as $CH_4$ dry reforming (Equation (3)), $CH_4$ steam reforming (Equation (4)), reverse-Boudouard (Equation (5)), and RWGS reactions (Equation (6)), which usually causes low $Y_{CH_4}$ [60,62].

$$CO_2 + CH_4 \rightleftharpoons 2CO + 2H_2 \quad \Delta H_{298K} = 247.3 \text{ kJ mol}^{-1} \tag{3}$$

$$CH_4 + H_2O \rightleftharpoons CO_2 + 3H_2 \quad \Delta H_{298K} = 206 \text{ kJ mol}^{-1} \tag{4}$$

$$CO_2 + C \rightleftharpoons 2CO \quad \Delta H_{298K} = 172.4 \text{ kJ mol}^{-1} \tag{5}$$

$$CO_2 + H_2 \rightleftharpoons CO + H_2O \quad \Delta H_{298K} = 41.2 \text{ kJ mol}^{-1} \tag{6}$$

The occurrence of these side reactions most likely was responsible for the observed steady and slow decrease in $X_{CO_2}$ at temperatures above the optimum value of 650 K, since $CO_2$ was consumed during most of these reactions while the decline in $Y_{CH_4}$ was visible. Moreover, the exothermic nature of the $CO_2$ methanation reaction and the likelihood of hot-spot formation makes it difficult to control the heat in the reactor. Razzaq et al. [63] reported that instead of the $Y_{CH_4}$ increasing at high temperatures, CO formation via RWGS reaction was favoured at temperatures above 673 K. Similarly, Panagiotopoulou et al. [64] observed that at reaction temperatures greater than 633 K, the formation rate of CO through RWGS was greater than the rate at which CO was consumed in the hydrogenation reaction, which in total led to the decrease in CO conversion.

The catalyst performance improved as the Ni loading increased until reaching optimal performance for a loading of 40% Ni. Any further increase in Ni loading reduced the catalyst activity for the methanation reaction. The initial increase in the catalyst's activity with Ni loading can be ascribed to more availability of active sites resulting from the

improved pore size as seen in the BET results (see Table 1) and the improvement in MSI as seen in their reducibility at lower temperatures (see Figure 5). Figure 8 reveals that the increase in Ni loading resulted in a decrease in CO selectivity up to 40% Ni loading. Studies have shown that Ni loading, together with $Ni^0$ particle size and morphology, influences the reaction pathways during $CO_2$ methanation owing to its role in the adsorption and activation of $H_2$ on the catalyst surface [65]. Small $Ni^0$ particle size will likely encourage a longer adsorption-dissociation route through which $CH_4$ or C is formed. The small $Ni^0$ particles may also have low coverage for $H_2$ dissociation which favours the desorption of CO from the surface without hydrogenation to $CH_4$ [44,66].

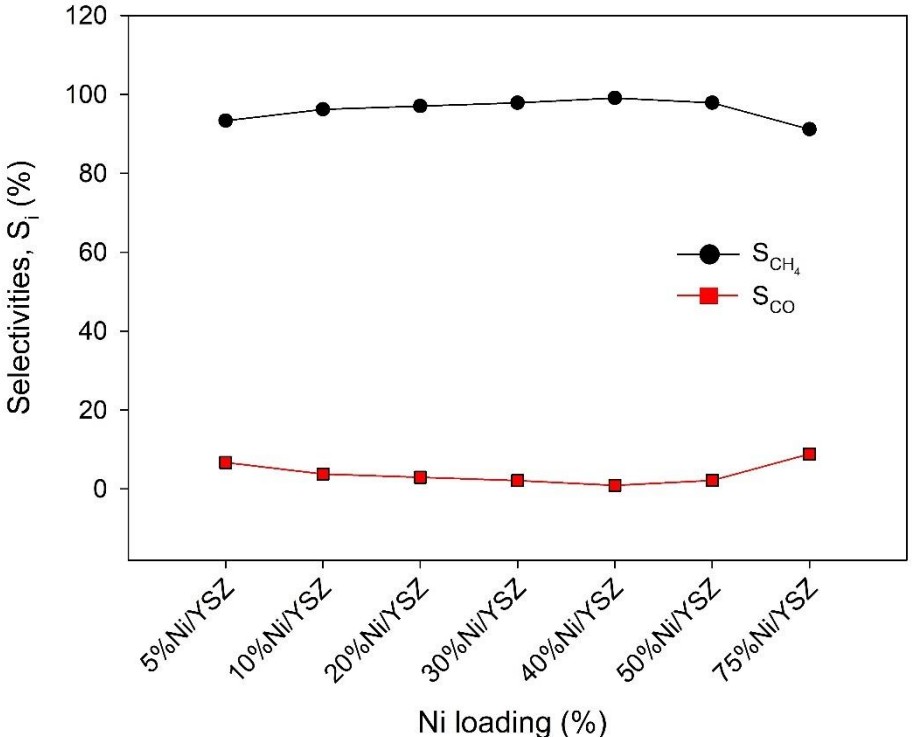

**Figure 8.** Activity test profile showing the selectivity to $CH_4$ and CO formation during $CO_2$ methanation at 643 K over Ni/YSZ catalysts. Reaction conditions: Total flow = 135 mL min$^{-1}$, $H_2$:$CO_2$ ratio = 4, pressure = 1 bar, GHSV 40,500 mL h$^{-1}$ g$^{-1}$cat.

Meanwhile, large $Ni^0$ particle size can influence the intermediate carbon species to be hydrogenated to $CH_4$ since there is a large coverage for $H_2$. This means more $CH_4$ production and less CO formation. Accordingly, the decrease in CO selectivity was observed as shown in Figure 8. can be attributed to the favoured reaction route towards the formation of $CH_4$ as the Ni content increases. Other studies have also shown that an increase in Ni loading encourages $H_2$ chemisorption and increased $H_2$ uptake, indicating high $H_2$ coverage on the catalyst surface [66,67]. However, too high Ni loading of over 40% affects the catalyst morphology reducing the overall catalyst activity.

From our results, the catalyst performance significantly declined after increasing the Ni% loading beyond 40%. This indicates that there was likely a change in the reaction path at higher Ni loading or inaccessibility of the reacting species to the active sites due to pore blockage. In any case, further study is required to determine the relationship between Ni loading and the number of active sites, as well as the optimum $Ni^0$ particle size that favours high $CH_4$ production and minimises or inhibits CO formation. Overall, the performance of the catalysts was in the order 40% Ni/YSZ > 30% Ni/YSZ > 20% Ni/YSZ > 50% Ni/YSZ > 10% Ni/YSZ > 5% Ni/YSZ > 75% Ni/YSZ at temperatures beyond 613 K. Therefore, the optimum catalyst choice was 40% Ni/YSZ. It is worth mentioning that this comparative study was carried out using an equal mass of catalysts. We also found that the catalyst

bed length in the reactor increased with higher Ni loading, indicating an increase in the volume and a decrease in the bulk density. This means that the Ni/YSZ catalysts at higher Ni loading would likely have a longer contact time for the reacting species to react than the catalysts at low Ni loading. The longer contact time may also explain why there was an improvement in the performance with higher Ni loading.

However, the catalyst activity dropped when the loading was increased beyond 40% Ni despite a favoured higher contact time. This is likely due to the enhanced clustering of the Ni particles as seen in the SEM images and the increase in the crystallite size at higher Ni loading (see Table 1).

## 3. Materials and Methods

### 3.1. Catalyst Synthesis

Ni-based yttria stabilised zirconia, Ni/YSZ was prepared by wetness impregnation of YSZ (YSZ: Tosoh-zirconia TZ-8YS, from Tosoh corporation, Tokyo, Japan) with an aqueous solution of nickel(II) nitrate hexahydrate ($Ni(NO_3)_2 \cdot 6H_2O$ Extra Pure, SLR, from Fisher Chemical™, Waltham, MA, USA). In a typical synthesis, the calculated weight of YSZ powder was added to an aqueous solution containing a calculated concentration of $Ni(NO_3)_2 \cdot 6H_2O$ (to achieve various Ni loading) and stirred at 500 rev min$^{-1}$ and 303 K temperature for 2 h. Thereafter, the temperature was raised to 323 K while the mixture remained under stirring until a slurry was formed. The mixture was then transferred to an oven where it was dried overnight at 353 K and calcined at 923 K for 3 h.

### 3.2. Catalyst Characterisation

The BET surface area of the samples was measured via $N_2$ adsorption using a Micromeritics® TriStar II Plus analyser at a temperature of 77.15 K. Before the analysis, the samples were degassed for 6 h at 373 K under vacuum. Powder X-ray diffraction of all the samples was carried out in a Proto Benchtop AXRD at 30.5 kV and 20.5 mA scanning from a 2-theta of 5° to 80° at an increment of 0.0149° using Cu-Kα with wavelength 1.5406 Å. An environmental Scanning Electron Microscope (SEM) Zeiss Evo10 and Tabletop electron microscope Hitachi TM3030 with an energy dispersive spectrometer (EDS) were used to take images and view the distribution of the various species on the catalyst surfaces through mapping. The reducibility of samples was determined through $H_2$-temperature programmed reduction (TPR) in a quartz tube reactor connected to an MKS Cirrus mass spectrometer.

### 3.3. Catalyst Testing

The performance of the synthesised catalysts for the $CO_2$ methanation was examined in a continuous flow quartz tube fixed-bed reactor (I.D. = 5.5 mm, wall thickness = 2 mm) at a temperature range of 473–703 K under atmospheric pressure. In a typical test, 200 mg of the catalyst was loaded into the reactor containing inert quartz wool at both ends of the catalyst. The reactor was horizontally positioned in a temperature-controlled furnace (Elite Thermal Systems. Ltd.: Model THH12/90/305, Market Harborough, UK). Before the reaction, a catalyst was first reduced by passing a stream of 20 mL min$^{-1}$ $H_2$ and 60 mL min$^{-1}$ $N_2$ through it at 923 K for 1.5 h. Subsequently, the reactor was cooled to the reaction temperature of 473 K and the catalyst was purged by passing a stream of $N_2$ (100 mL min$^{-1}$) for 5 min. Thereafter, a gaseous feed stream containing $N_2$ (60 mL min$^{-1}$), $H_2$ (60 mL min$^{-1}$) and $CO_2$ (15 mL min$^{-1}$) was introduced at 473 K under atmospheric pressure and the temperature was increased gradually up to 693 K. The reactant flow rate was controlled using calibrated mass flow controllers. The product of the reaction was channelled through a drying system before being sent to a gas chromatograph (GC) Shimadzu GC-2014 for analysis. The GC, which was equipped with a thermal conductivity detector and a column (ShimCarbon ST, length 200 m, inner diameter 0.35 mm), was

operated under argon as the carrier gas. The catalyst activities were estimated by applying Equations (7)–(9) [68], respectively:

$$X_{CO_2} = \left( \frac{F_{CO_{2,in}} - F_{CO_{2,out}}}{F_{CO_{2,in}}} \right) \times 100\% \tag{7}$$

$$S_i = \left( \frac{F_{i,out}}{F_{CH_{4,out}} + F_{CO,out}} \right) \times 100\% \tag{8}$$

$$Y_{CH_4} = \left( \frac{X_{CO_2} \times S_{CH_4}}{100} \right) \% \tag{9}$$

where $X_{CO_2}$ = CO$_2$ conversion, $S_i$ = Selectivity of product i (CO or CH$_4$), $Y_{CH_4}$ = CH$_4$ yield, $F_{CO_{2,in}}$ = molar flowrate of inlet CO$_2$ in mol/s, $F_{CO_{2,out}}$ = flowrate of unreacted CO$_2$ in product and $F_{CH_{4,out}}$ is the molar flowrate of CH$_4$ in the product, $F_{i,out}$ is the molar flowrate of CH$_4$ or CO in the product.

## 4. Conclusions

The effect of Ni loading on the catalytic activity of Ni/YSZ catalysts during CO$_2$ methanation was studied. The Ni/YSZ catalysts were prepared by the wetness impregnation method with variations in the amount of Ni content. The N$_2$ adsorption/desorption isotherms of these catalysts revealed that they were formed in a type IV isotherm with an H3-type hysteresis loop which confirms the presence of mesopores and micropores in their structures. The similarity in the isotherms of all the catalysts, regardless of the amount of Ni content, suggests that the YSZ structure was preserved after the catalyst preparation. XRD results indicated that higher Ni loading favoured the formation of larger Ni particle sizes as obtained from the Scherrer equation. The EDX mapping of these catalysts further revealed the presence of more clusters at higher Ni loading. The reducibility test showed that the increase in Ni loading caused the reduction temperature to shift to the right. This suggests that a higher temperature is required to reduce the catalysts with higher Ni loading as a result of strong MSI. An activity test of these catalysts at different temperatures from 473 to 663 K was carried out. The results showed that both CO$_2$ conversion and CH$_4$ yield were favoured with an increase in temperature until an optimum temperature of around 613 to 653 K, beyond which there was a decline in the activity. The decrease in activity at the elevated temperatures beyond the optimum value was ascribed to the occurrence of side reactions, which are favoured at higher temperatures. It was also found that the catalyst performance did not increase proportionately with Ni loading. The amount of Ni in the catalyst was increased from 5% to 75% and the optimum loading was found to be between 30% and 40% Ni loading. The improved catalyst performance with higher Ni loading was ascribed to the increase in coverage for H$_2$ adsorption and activation. We also noted that the bed length was longer for the catalysts with higher Ni loading and this encouraged higher contact time. However, the catalyst activity dropped as the Ni loading increased beyond 40% Ni. Therefore, further study is recommended to determine how the variation in Ni loading influences the number of active sites. This will help to fully understand the mechanism of the reactions during CO$_2$ methanation over Ni/YSZ at different Ni loadings.

**Author Contributions:** Conceptualization: O.O., A.E.-k. and R.S.-W.; writing—original draft: O.O.; investigation: O.O. and A.J.M.; methodology: A.J.M. and A.E.-k.; formal analysis: A.J.M. and A.E.-k.; data curation: A.J.M.; validation: A.E.-k.; supervision: R.S.-W.; writing—review & editing: R.S.-W. All authors have read and agreed to the published version of the manuscript.

**Funding:** This research was funded by Petroleum Technology Development Fund (grant number: 18UK/PHD/025).

**Institutional Review Board Statement:** Not applicable.

**Informed Consent Statement:** Not applicable.

**Data Availability Statement:** Not applicable.

**Acknowledgments:** The authors acknowledged financial support provided by the Petroleum Technology Development Fund (grant number: 18UK/PHD/025) for undertaking this current study.

**Conflicts of Interest:** The authors declare no conflict of interest.

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
