# Peer review of "Investigating the Effect of Ni Loading on the Performance of Yttria-Stabilised Zirconia Supported Ni Catalyst during CO2 Methanation"

_methane, doi:10.3390/methane2010007_

Round 1

Reviewer 1 Report

The process that the authors propose; which is using the renewable hydrogen for Sabatier process for the production of SNG is a thermodynamically favorable process. However, this is not economically viable. The market price of natural gas is low

How was the reduction temperature decided. 

There is something wrong with Fig. 1.

Throughout the manuscript, the Figure numbers are not referenced properly. Everywhere it shows error. It looks like the authors did not even bother to read through the system built pdf before submission. 

In the discussion of SEM images, the authors mention that there is clustering of Ni particles with increase in the Ni loading, however, its difficult to make out the clustering from the SEM images.

What are the dark spots in the EDS ? The legend does not mention what the dark spots are. 

I fail to understand why the TPR is plotted as a function of time!!. Generally TPR is plotted as a function of temperature. I completely fail to read this graph. Please replot this with temperature as X axis so that the readers can understand. 

Please correct Fig. 6. This does not make any sense. I can’t make out what is what!!

Please compare the experimental measurements with equilibrium predictions. CO2 methanation reaction is limited by thermodynamics at temperatures around 400 K. In fact higher conversion is thermodynamically possible at lower temperatures, however, its kinetically limited. So please plot the equilibrium conversion line in Fig. 6

The authors mention that Ni particle size influences the adsorption of H2 and CO2. You also cite a reference for that. I request you to do some more literature search on this. There are several studies that says that CO2 gets activated on the support. It needs a basic site for adsorption. 

The discussion of coverage on small and large particle size of Ni is not really convincing. The particle size that you report is from 20-40 nm. It’s quite large compared to the size of H atoms. Furthermore, if you want to make a claim that large Ni particles promote hydrogenation of C to CH4, you need a proof for that. 

Author Response

Reviewer 1

Comment 1:

The process that the authors propose; which is using the renewable hydrogen for Sabatier process for the production of SNG is a thermodynamically favorable process. However, this is not economically viable. The market price of natural gas is low.

Response

Thank you for pointing out the economical concern of this process. CO2 methanation mainly aims to mitigate CO2 emission into the atmosphere, thereby, reducing the effect of climate change. This statement “Apart from the climate mitigation point of view, it is also believed that CO2 methanation will be economically competitive if the electrolytic process for the required H2 is further improved and reduced in cost. As electricity from renewable energy sources has become the cheapest type of electricity in many parts of the world, and with the current developments with natural gas prices, favourable economic conditions are visible.” has been included in the introduction.

Comment 2:

How was the reduction temperature decided. 

Response

A clarification on how the reduction temperature was selected was included in the text.

Comment 3:

There is something wrong with Fig. 1.

Response

We used the original SigmaPlot graphs in the text which probably resulted in problems during pdf generation in the Journal system. In this regard, all the figures in this manuscript have been converted into JPEG format.

Comment 4:

Throughout the manuscript, the Figure numbers are not referenced properly. Everywhere it shows error. It looks like the authors did not even bother to read through the system built pdf before submission. 

Response

We apologise for this oversight. The references were corrected, and all hyperlinks from the figures have been removed.

Comment 5:

In the discussion of SEM images, the authors mention that there is clustering of Ni particles with increase in the Ni loading, however, its difficult to make out the clustering from the SEM images.

Response

Thank you for this observation as this was not clearly stated in the manuscript. We have now added a statement to explain better the SEM scans that identify the presence of Ni clustering.

Comment 6:

What are the dark spots in the EDS ? The legend does not mention what the dark spots are. 

Response

Some explanation was added to the text. The dark spots seen in most of the EDX mapping of the catalysts are due to the cavities resulting from their rough surfaces.

Comment 7:

I fail to understand why the TPR is plotted as a function of time!!. Generally TPR is plotted as a function of temperature. I completely fail to read this graph. Please replot this with temperature as X axis so that the readers can understand. 

Response

Thank you for your comment. We have replotted the H2-TPR profile with the temperature placed on the X-axis. 

Comment 8:

Please correct Fig. 6. This does not make any sense. I can’t make out what is what!!

Response

Thank you for this observation. The plots have been corrected to make them easier to read.

Comment 9:

Please compare the experimental measurements with equilibrium predictions. CO2 methanation reaction is limited by thermodynamics at temperatures around 400 K. In fact higher conversion is thermodynamically possible at lower temperatures, however, its kinetically limited. So please plot the equilibrium conversion line in Fig. 6

Response

The CO2 methanation reaction is thermodynamically feasible at low temperatures, but it has been reported in the literature that high operating pressure will be required to conduct CO2 methanation at low temperatures for a favourable equilibrium composition. Some clarification was added to the text.

Comment 10:

The authors mention that Ni particle size influences the adsorption of H2 and CO2. You also cite a reference for that. I request you to do some more literature search on this. There are several studies that says that CO2 gets activated on the support. It needs a basic site for adsorption. 

Response

Thank you for this observation. It is true that CO2 activation mainly occurs on the support since it needs a basic site. The statement has been corrected. Some new references were added to support this observation.

Comment 11:

The discussion of coverage on small and large particle size of Ni is not really convincing. The particle size that you report is from 20-40 nm. It’s quite large compared to the size of H atoms. Furthermore, if you want to make a claim that large Ni particles promote hydrogenation of C to CH4, you need a proof for that.

Response

Some correction was added to the text to clarify this. Many reports confirmed that small Ni0 particles favour CO formation and larger Ni0 particles promote CH4 formation due to large H2 coverage. See [67] or [44]. This is up to a certain level – too high Ni loading (over 40% in our case) affects catalyst morphology reducing overall catalyst activity

Reviewer 2 Report

The manuscript is nicely written and can be accepted for publication. 

However, giving the conclusion part in a point by point basis thereby highlighting the main outcomes of this research will be very helpful for the readers. 

Author Response

Reviewer 2

The manuscript is nicely written and can be accepted for publication. 

However, giving the conclusion part in a point by point basis thereby highlighting the main outcomes of this research will be very helpful for the readers.

Response

This point is well noted. The conclusion has been modified.

Reviewer 3 Report

1. Please check and improve “Error! Reference source not found.” in line 209, 217, 237, 246, 254, etc.

2. Figure 1, the adsorption/desorption curves show incomplete, such as the “5%Ni/YSZ, 10%Ni/YSZ, 20%Ni/YSZ, please check and improve.

3. The lines in Figure 6 show incomplete, please check and improve.

4. The meaning of the two lines in Figure 7 is not clearly marked.

5. Please refine “Conclusion”.

Author Response

Reviewer 3

  1. Please check and improve “Error! Reference source not found.” in line 209, 217, 237, 246, 254, etc.

Response

Thank you for this observation. These errors have been resolved.

  1. Figure 1, the adsorption/desorption curves show incomplete, such as the “5%Ni/YSZ, 10%Ni/YSZ, 20%Ni/YSZ, please check and improve.

And

  1. The lines in Figure 6 show incomplete, please check and improve.

Response

The figures were plotted with SigmaPlot and the original files were added to the text which created some problems during pdf generation by the platform. Therefore, we have converted all the Figures into Jpeg format.

  1. The meaning of the two lines in Figure 7 is not clearly marked.

Response: This has been corrected

  1. Please refine “Conclusion”.

Response

This point is well noted. The conclusion has been modified.

Round 2

Reviewer 1 Report

The footnote for Table 2 is wrong. Instead of Fig. 4 it should be Fig. 5